# The fungus *Leptosphaerulina* persists in *Anopheles gambiae* and induces melanization

**Godfrey Nattoh**[1,2]*, **Joel L. Bargul**[1,3], **Gabriel Magoma**[2,3], **Lilian Mbaisi**[1],
**Hellen Butungi**[1,4], **Enock Mararo**[1], **Evan Teal**[1], **Jeremy Keith Herren**[1]

**1** International Centre of Insect Physiology and Ecology, Nairobi, Kenya, **2** Pan African University Institute for Basic Sciences Technology and Innovation, Nairobi, Kenya, **3** Department of Biochemistry, Jomo Kenyatta University of Science and Technology, Nairobi, Kenya, **4** Wits Research Institute for Malaria, University of the Witwatersrand, Johannesburg, South Africa

* gindinda@icipe.org

**Data Availability Statement:** Sequences obtained in this study have been deposited in the GenBank database under the following accession numbers: MT437213-MT437220; MT433814; MT435649.

## Abstract

*Anopheles* mosquitoes are colonized by diverse microorganisms that may impact on host biology and vectorial capacity. Eukaryotic symbionts such as fungi have been isolated from *Anopheles*, but whether they are stably associated with mosquitoes and transmitted transstadially across mosquito life stages or to subsequent generations remains largely unexplored. Here, we show that a *Leptosphaerulina* sp. fungus isolated from the midgut of *An. gambiae* can be stably associated with *An. gambiae* host and that it imposes low fitness cost when re-introduced through co-feeding. This fungus is transstadially transmitted across *An. gambiae* developmental stages and to their progeny. It is present in field-caught larvae and adult mosquitoes at moderate levels across geographical regions. We observed that *Leptosphaerulina* sp. induces a distinctive melanotic phenotype across the developmental stages of mosquito. As a eukaryotic symbiont that is stably associated with *An. gambiae* *Leptosphaerulina* sp. can be explored for paratransgenesis.

## Introduction

Management of the *Anopheles gambiae* species complex, which includes primary vectors of malaria, a parasitic disease with devastating consequences especially in the sub-Saharan Africa [1, 2], is facing immense challenges arising from emergence of insecticide resistance strains and biting outdoors behaviors, characteristics associated with changes in the vector biology [2–5]. The realization that these vectors associate with a consortium of gut microbes, and that these could be having implications on the host biology has motivated research on diverse mosquito-associated microorganisms. Mosquitoes are known to harbor bacteria, viruses, and eukaryotes (e.g. fungi), all of which have been investigated in the context of host-symbiont interactions with the hope of identifying candidates that stably co-exist and influence mosquitoes host physiology [6–8]. The recent proof-of-concept on the development of non-chemical approaches using *Wolbachia* symbiont that involve release of *Aedes* infected with *Wolbachia* with the aim of replacing wild *Aedes* populations with laboratory colonies over years to reduce transmission of dengue and zika, signify the prospects of exploiting isolates with ideal

**Funding:** G.N. was supported by African Union under the auspice of Pan African University, Institute for Basic Sciences Technology & Innovation (PAUSTI) Postgraduate Scholarship. We also gratefully acknowledge icipe core funding provided by UK Aid from the Government of the United Kingdom, Department for Inter- national Development (DFID); the UK, Swedish International Development Cooperation Agency (Sida); the Swiss Agency for Development and Cooperation (SDC); Federal Ministry for Economic Cooperation and Development (BMZ), Germany; Federal Democratic Republic of Ethiopia; and the Kenyan Government. JKH was funded by the Wellcome Trust [107372] while JLB was supported by DELTAS Africa Initiative grant # DEL-15-011 to THRiVE-2. The DELTAS Africa Initiative is an independent funding scheme of the African Academy of Sciences (AAS)'s Alliance for Accelerating Excellence in Science in Africa (AESA) and supported by the New Partnership for Africa's Development Planning and Coordinating Agency (NEPAD Agency) with funding from the Welcome Trust grant # 107742/Z/15/Z and the UK government. The views expressed herein do not necessarily reflect the official opinion of the donors. The funders had no role in study design, data collection and analysis, decision to publish, or preparation of the manuscript.

**Competing interests:** The authors have declared that no competing interests exist.

microbe-vector interactions for disease management [7–9]. *Anopheles gambiae* that were microinjected with *Wolbachia* were partially protected against *Plasmodium falciparum* and field survey of *Wolbachia* infection status on this host indicate low infection intensities [10], and this has led to a suggestion that wild caught *Anopheles gambiae* with *Wolbachia* could have acquired these symbiont from plant and ectoparasitic mites or midges [11]. A number of studies have isolated bacteria from *Anopheles* mosquitoes and shown that they influence host development, nutrition, and association with pathogens [12–16]. Fungi have been isolated from anopheline mosquitoes [17–20], and previous studies indicate that naturally occurring *Penicillium chrysogenium* enhances *Anopheles* susceptibility to *Plasmodium* infection [21], while *Talaromyces* sp. in *Aedes aegypti* makes this host less permissive to dengue viruses [22]. It will be important to understand whether fungal symbionts are pervasive and persistent in *Anopheles*, which could have implications for disease transmission. Studies on non-entomo-pathogenic fungi isolated from *Aedes* mosquitoes indicate that yeasts form the core mycobiota and that diverse fungi infect more than 70% of the crop and gut of these mosquitoes, suggesting nutritional benefits to the host [23–26]. Establishing ways of sustainable dissemination of the recently identified uncultivable microsporidian symbiont in *An. arabiensis* to increase its prevalence in the wild could translate into reduced malaria cases [27]. Identification of heritable cultivable eukaryotic fungal symbionts [mycobiomes] of anopheline mosquitoes and understanding their interaction with these hosts could inform selection of ideal symbiont candidates for the delivery of anti-plasmodium molecules in a paratransgenic transmission blocking approach to control malaria.

It is not clear whether cultivable fungi previously isolated from anopheline mosquitoes were stage specific or pervasive across developmental stages and conferred harmful or beneficial effects on host. A rich diverse fungal symbiont were observed in the gut of adult *Aedes* mosquitoes suggesting that similar factors shaping composition of bacteria in these host could also be influencing mycobiomes [23]. Several families of *Culicidae* have been shown to co-exists with members of filamentous fungi [17], though their relationship with the host remains unclear. Pathogenic fungi such as *Candida parapsilosis* isolated from various developmental stages of the wild *Anopheles* sp., *Aedes* sp., *and Culex* sp. signify the possible role these vectors play in disseminating some pathogenic fungi [20]. Functional studies indicate that the *Anopheles gambiae* midgut is inhabited by ascomycete fungi such as *Penicillium chrysogenum* and *Talaromyces* species that confer modulation of malaria parasite and dengue infection respectively, thereby rendering the host more susceptible to these infections [21, 22]. It is possible that infection by these fungi could have caused an increase in host susceptibility to these pathogens hence the need to study host-fungal symbiont interaction and the fitness cost thereof. An ideal fungal symbiont would possess properties such as stable association with host stages and conferring minimal influence on the host biology [8]. These underpinnings are important to inform a paratransgenic approach that is appropriate for sustainable utilization of novel symbionts to control vector-borne diseases. Modulation of mosquito immune system through host production of effector molecules with anti-parasitic properties is a plausible mechanisms utilized by most symbionts to bring about these desired properties [15, 16, 28].

Field-collected mosquitoes have highly diverse microbial taxa, which suggest the important role played by the environment in shaping microbial composition [12, 29, 30]. Most anopheline lose microbes at pupae stage as they shed their contents while emerging to adults and subsequently acquire additional symbionts from their environment, a phenomenon that was described as microbial reduction [31]. It is not clear to what extent fungal symbionts can evade this microbial reduction to persist in these vectors and influence the hosts' biological fitness across developmental life stages. Whether adult stages acquire fungal microbes exclusively from their environment or through transstadial transmission from immature stages remains

unresolved [12, 13, 32]. Non-entomo-pathogenic fungal symbionts undergoing transmission along the mosquito developmental stages could be useful as potential candidates for paratransgenesis [33].

In the present study, we isolated and identified several fungal isolates that interact with the main malaria vector, *Anopheles gambiae*. One such fungus was found to have a low level of pathogenicity and was identified as *Leptosphaerulina* sp. This *Leptosphaerulina* sp. seemed to activate the host immune system to produce deposits of melanin (melanization) on the fat body [34]. Melanosis has been reported in *Anopheles gambiae* infected with *Elizabethkingia meningoseptica* and this interaction was reported to influence vertical transmission [35]. The induction of melanosis by *Leptosphaerulina* sp. is believed to help in the establishment and stable inheritance across developmental phases, suggesting that it could be transmitted either horizontally and/or vertically.

## Results

### *Leptosphaerulina* sp. stably associates with *Anopheles gambiae* life stages and confers minimal effects on the host

Using standard culture-based techniques, nine distinct fungal isolates were recovered from the midgut of female *Anopheles* sp. maintained in the semi-field insectaries at *icipe*-Thomas Odhiambo Centre, western Kenya, and identified using the rRNA internal transcribed spacer [ITS] region [36]. Sequence analysis revealed that isolates were similar to ascomycetes fungi (Table 1). To assess whether these fungi could be reintroduced and maintained in the laboratory-reared *Anopheles gambiae*, we prepared and inoculated a one-off dose of 60mg [approximately $3.9 \times 10^3 – 5.5 \times 10^5$ spores/mL] whole ground spores, and mycelium fungus into the rearing water with uninfected 1$^{st}$ instars larvae [L1 stage] (graphically illustrated in S1A Fig). Using fungus specific primers (Lepto 379f/566r and Lepto 521f/896r developed from whole genome fungus sequences) and ITS1/ITS4 we observed that one isolate namely *Leptosphaerulina* sp. (here referred to as Lepto) had infected 90.92% of L3/L4 instar larvae at high densities (Kruskal Wallis Test, $p = 0.001$; Fig 1A), and 7-day old adult stage (Kruskal Wallis Test, $p < 0.0001$; Fig 1B) as manifested by the densities of infection. Prior to fungal introduction, cleaning of mosquito eggs was undertaken to establish fungus-free colonies (S1B Fig, **panel W-Z**). Pupae water was also confirmed to be fungus-free using qPCR and culture methods. The presence of *Leptosphaerulina* sp. in adults signified persistence, which is indicative of effective

**Table 1. Fungal isolates, their ITS gene sequences and related taxas from GenBank and fungal database, and their origin from digestive tracts of *Anopheles gambiae*s. l.**

| Isolate Code (GenBank Accession no.) | Related Taxa (GenBank Accession no.) | BlastN (%) | Taxanomic division Phylum | Compartment isolated |
|---|---|---|---|---|
| **Isolate 2(MT437213)** | *Hyphopichia burtonii* (KY103604) | 97.3 | Ascomycota | Ovary |
| **Isolate 4(MT437214)** | *Hyphopichia* sp. (KY103604) | 93.7 | Ascomycota | Ovary/midgut |
| **Isolate 5(MT437215)** | *Penicillium georgiense* (NR_121325) | 96.9 | Ascomycota | Pupae |
| **Isolate 11(MT437216)** | *Periconia* sp. (LN813031) | 99.4 | Ascomycota | Gut/ovary |
| **Isolate 13B(MT433814)** | *Leptosphaerulina chartarum* (KJ863505) | 99.1 | Ascomycota | Ovary/Midgut |
| **Isolate 23(MT437217)** | *Cladosporium cladosporioide*(KF159973) | 99.6 | Ascomycota | Whole mosquito |
| **Isolate 30(MT437218)** | *Hasegawazyma lactosa*(FJ515208) | 77.1 | Basidiomycota | Larvae |
| **Isolate 55(MT437220)** | *Epicoccum* sp.(KX965725) | 99.4 | Ascomycota | Gut/ovary |
| **Isolate 50(MT437219)** | *Alternaria alternata*(MF099865) | 99.4 | Ascomycota | Larvae |
| **Isolate 58(MT435649)** | *Lichtheimia hyalospora*(NR_111440) | 97.5 | Ascomycota | Gut/Ovary |

Accession numbers are given in brackets.

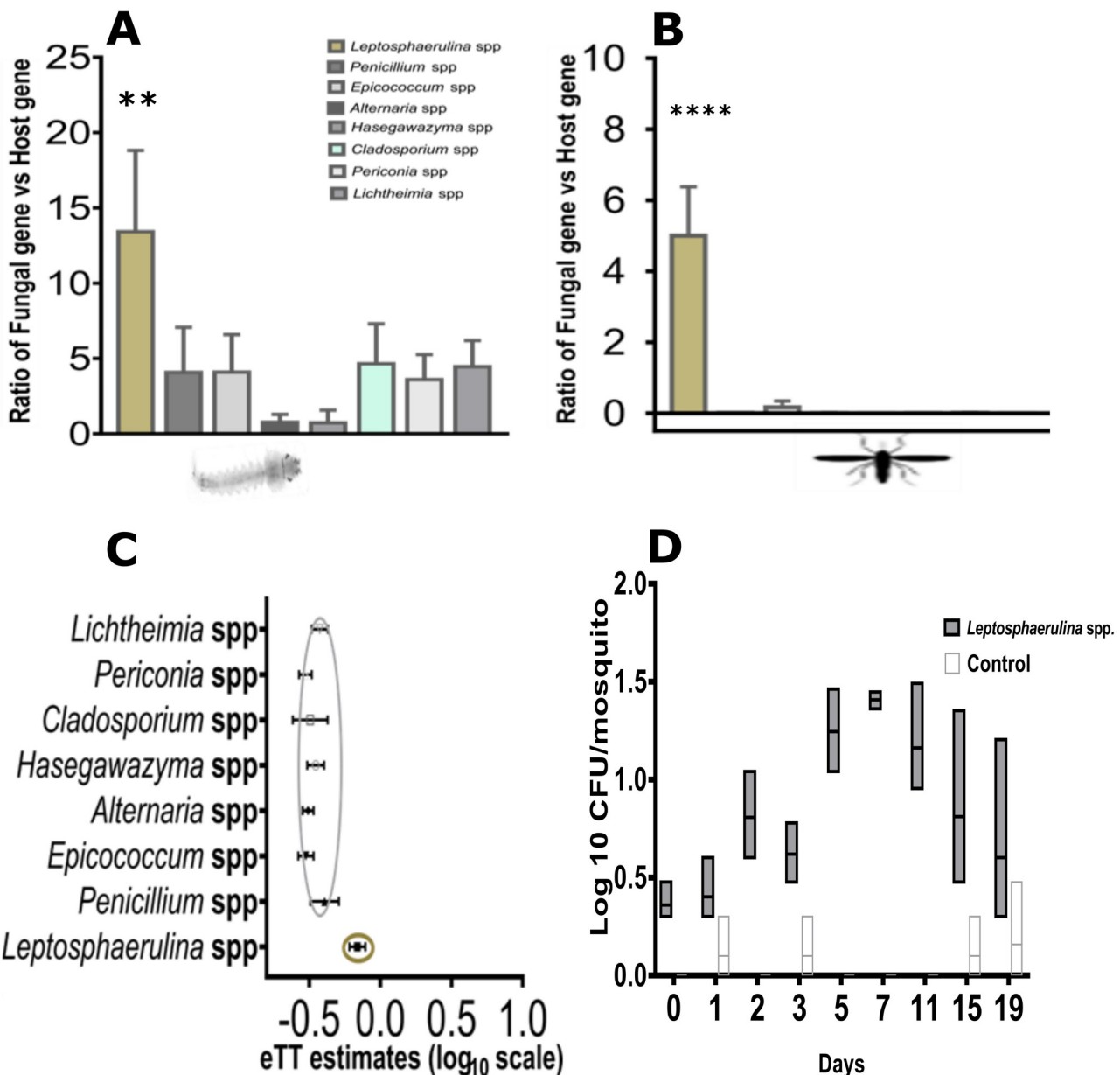

**Fig 1. Persistence and transmission of *Leptosphaerulina* sp. fungus isolate across developmental stages.** Individual fungi were inoculated to uninfected 1st instar larvae stages and the presence and densities expressed as ratio of fungal gene (Lepto521F/896R) and Ribosomal gene (S7), were tested using qPCR-HRM assay at larval (L3/L4) and four-day old adult stages. (**A**) Density of *Leptosphaerulina* sp. was found to persist in 90.92% of surviving larvae (N = 128, *p = 0.001*) and 66.43% of emerging adult stages (**B**; *p = <0.0001*). These were confirmed by culture re-isolation and re-sequencing methods. (**C**) The effective transstadial transmission (eTT estimates) based on transformed log differences between densities of infection between larvae and adult denotes that *Leptosphaerulina* sp. persisted along developmental stages. (**D**) These were confirmed by monitoring the isolate in female for 19 days by culture and enumerating fungal CFUs on Sabouraud agar laced with 1% HCl and tetracycline. The mean bar represents mean ±95%CI, while ** represent *p <0.001*, **** represent *p <0.00001*.

transstadial transmission from larvae to adult (Fig 1C). To address whether *Leptosphaerulina* sp. could colonize anopheline midgut, emerging adults that were infected at L1 instar larvae were observed to retain infection when monitored by fungus colony forming units for 19 days (Fig 1D). *Leptosphaerulina* sp. was detected for the entire monitoring period. To confirm this transmission, we re-isolated and re-sequenced the fungus from the 10-day old adults.

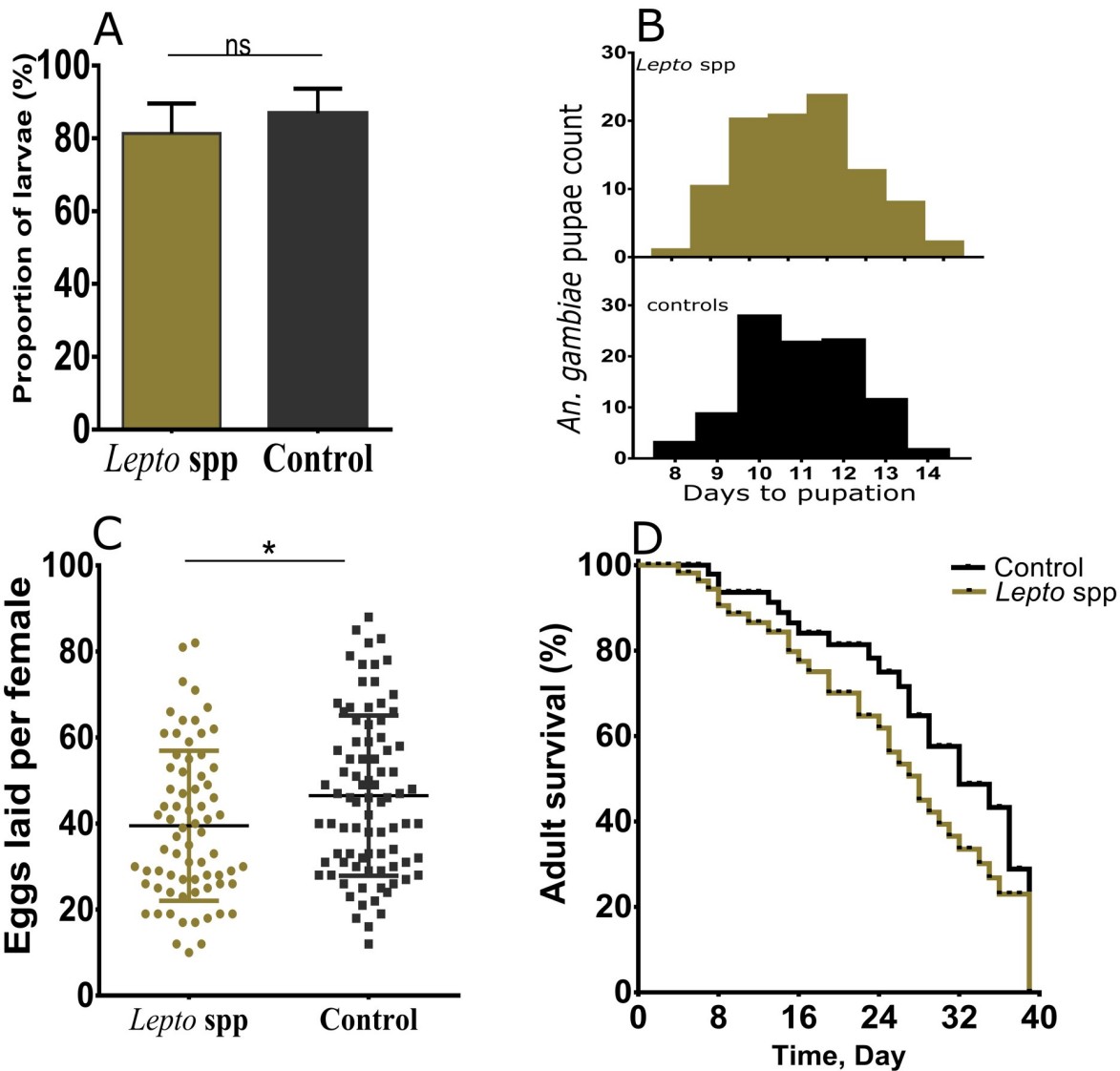

**Fig 2. Effects of *Leptosphaerulina* sp. (*Lepto* sp.) on fitness of *An. gambiae*. A)** The proportion of larvae pupating upon infection with fungus indicate that the isolate did not impose deleterious effect on their development (Mann-Whitney U = 24.5, *p* values = 0.1675). **B)** The duration taken by fungus infected larvae to pupate was significantly delayed (Mann-Whitney U = 15837, *p* = 0.0161). **C)** A reduction in the number of eggs laid by individuals females infected with fungus was noted (Mann-Whitney U = 2758, *p* = 0.047). **D)** Adult survival did not differ significantly between fungus infected and uninfected mosquitoes lines (two-sided log-rank Mantel–Cox, $\chi 2$ = 0.6855, df = 1, *p* = 0.4077). Infection statuses were confirmed *post hoc* using qPCR-HRM assay. The mean bar represents mean±S.E.M., while * represent p <0.05, ns = not significant (*p*>0.05).

Microscopic evaluation of the isolate indicates that *Leptosphaerulina* sp. forms pseudo hyphae, septae, and spores (S2 Fig). We hypothesized that this isolate could be conferring beneficial effects on the physiology of *Anopheles gambiae*, the primary vector for malaria.

Mosquitoes that ingested fungus developed melanotic phenotypes persisting across all developmental stages, but the occurrence of melanin were not lethal to reduce the proportion of infected larvae (Mann-Whitney U = 24.5, *p* values = 0.1675; Fig 2A). Notably, the infected larvae delayed to emerge and had reduced emergence rate (Mann Whitney U = 15837,

$p<0.016$; Lepto; n = 172, mean = 11.384±0.119 days, control: n = 214, mean = 10.967±0.092 days; Fig 2B). Analysis of egg counts from females showed a significant reduction in the fungus infected groups (Mann Whitney U = 2758, $p$ = 0.0047; Fig 2C). Using Log-rank test (Mantel-Cox), we did not observe any significant effect on the survival of mosquitoes harboring *Leptosphaerulina* sp. ($x^2$ = 0.655, df = 1, $p$ = 0.4077; Fig 2D). Infection statuses were confirmed *post hoc* using qPCR. Taken together, we conclude that *Leptosphaerulina* sp. is virulent and has effects on host fitness especially on the proportions of surviving larvae and adult longevity. Besides, we established that 'screen-house' colonies maintained at our Centre harbored moderate levels of *Leptosphaerulina* sp. and this led to the hypothesis that mosquitoes naturally acquired and maintained this fungus. Consequently, we evaluated the presence and densities of *Leptosphaerulina* sp. across different geographical locations in Kenya. We observed that *Leptosphaerulina* sp. could be detected in the field-caught mosquitoes from a wide geographical locations in Kenya denoting the occurrence of natural infection (prevalence ~ 10%; Table 2), and fungus densities in these natural infection remaining relatively constant at larvae and adult stages (Mann Whitney test, $p$ = 0.491; S3 Fig).

## Vertical transmission of *Leptosphaerulina* sp. in *Anopheles gambiae* is dependent on maternal infection levels

To explore whether *Leptosphaerulina* sp. can be inherited by the offspring derived from infected colonies, infected males and females were separately sexed and mated with uninfected virgins of the opposite sexes. Illustrations of mating combinations utilized are shown in Fig 3A. We observed high prevalence of infected offspring (~73%) when both parents were infected with *Leptosphaerulina* sp. at L1 (Kruskal Wallis Test, $p< 0.0001$; Fig 3B). In contrast, 61% of the offspring acquired infection when mothers were infected relative to 49% when male mosquitoes had acquired fungus and transferred to females before ovipositing ($p$ = 0.606, Pearson's chi-squared test, n = 12). We observed that the presence of *Leptosphaerulina* sp. did not affect fecundity because larvae hatching from the F1 infected females from all mating combinations confirmed *post hoc* did not vary significantly (Kruskal Wallis Test, $p$ = 0.291;S4A Fig). We also determined fungus load in these progeny by assaying densities of fungi in specific compartments of the female progeny. Female progeny were used because they were more likely to acquire and retain infection based on higher infection levels in female whole mosquito compared to their male counterpart (Mann Whitney test, $p$ = 0.0003; S4B Fig). Tissue dissections of the individual midgut and reproductive tissues depicted a persistent colonization of the female midgut (Kruskal Wallis Test, $p$ = 0.0056; S4C Fig). Comparison of midgut densities between Go and their offspring indicate that maternal infection influences the amount of fungus acquired by the offspring ($r^2$ = 0.1008, $p<0.05$; Fig 3C). Maternal transmission occurred with greater efficiency than paternal transmission, as determined by comparing densities of *Leptosphaerulina* sp. in offspring infection from maternal and paternal modes of inheritance (Mann Whitney test, $p$ = 0.0213; Fig 3D).

**Table 2. The prevalence of candidate fungal species in Anopheles gambiae sl. mosquito vectors across geographically dispersed regions of Kenya.**

|  | Central Kenya | | Western Kenya | | | Coastal Kenya | |
|---|---|---|---|---|---|---|---|
|  | Mwea (n = 305) | Huruma (n = 70) | Nyawira (n = 149) | Kirindo (n = 93) | Ahero (n = 429) | Kilifi (n = 97) | Malindi (n = 102) |
| Prevalence of Lepto (%) | 22.62 | 27.14 | 18.12 | 12.9 | 16.55 | 18.56 | 11.76 |

Lepto species naturally co-exist with major malaria vector are moderate prevalence.

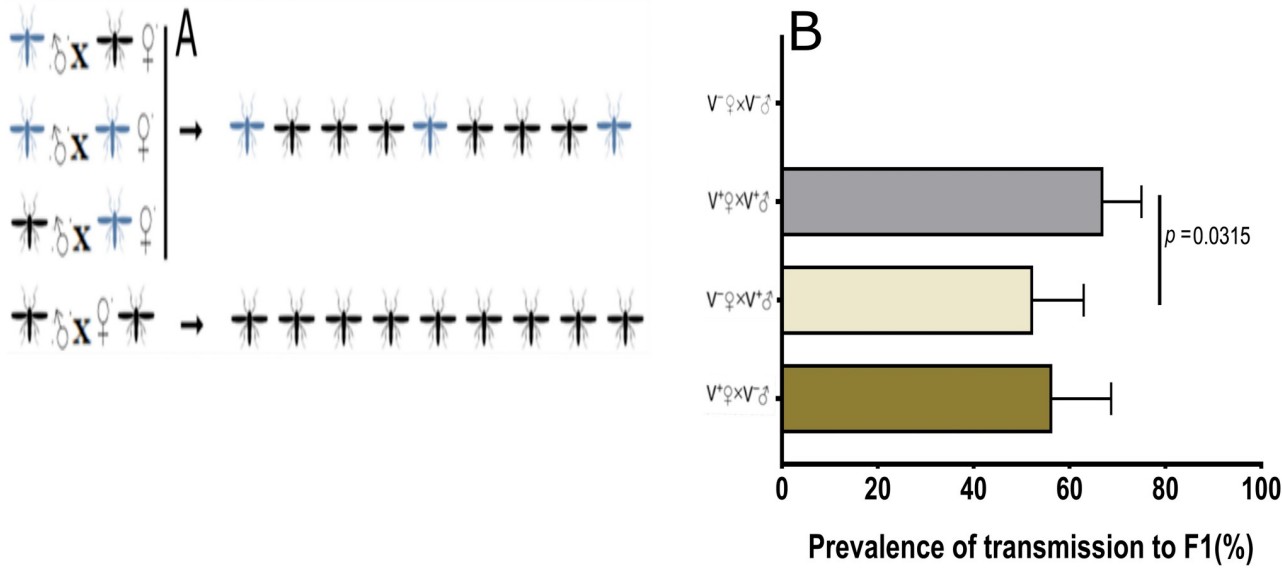

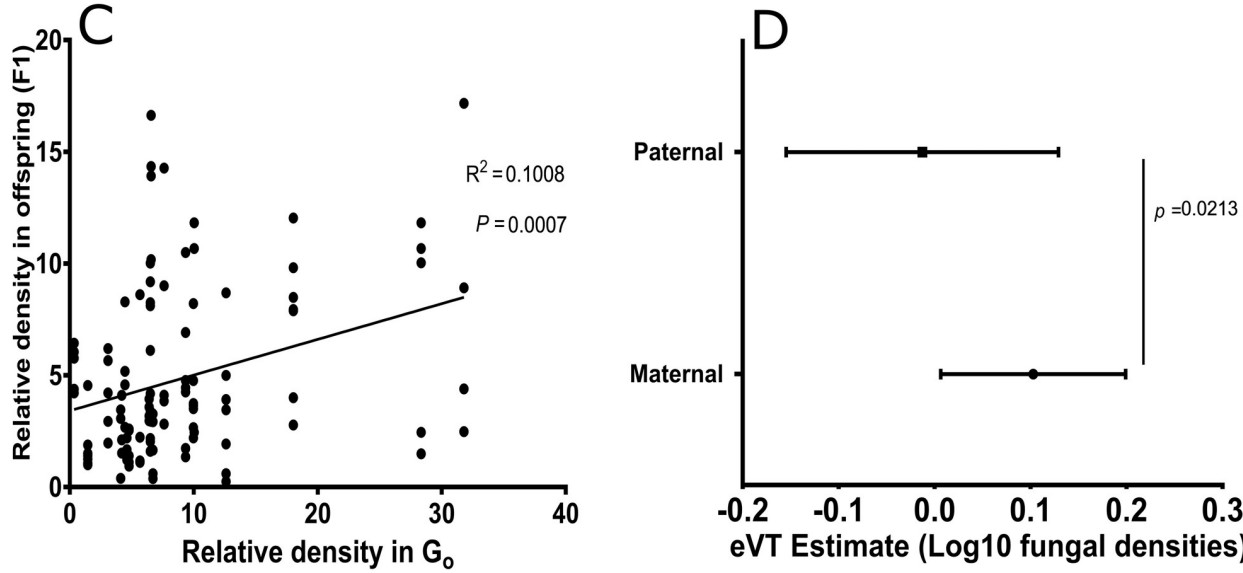

**Fig 3. Generational transmission of *Leptosphaerulina sp.* and effects on offspring of *An. gambiae*. A**) The graphics illustrate reproductive manipulation utilized in the experiment, black *Anopheles* signify wild type adults (uninfected) mated with adults exposed to fungi through co-feeding at 1st instars larvae stage (in blue) with their corresponding offspring indicated by an arrow. **B**) Mating infected female with uninfected males ($V^+♀xV^-♂$) or mating combination when both sexes have infection ($V^+♀xV^+♂$) resulted in 61.2% and 72.6% of offspring respectively acquiring infection. The offspring arising from mating combinations when both sexes were infected acquired higher infections than their counterparts from infected males ($p = 0.0315$). **C**) In a linear correlation, the densities of fungal infections in parents influenced transmitted densities in offspring ($r^2 = 0.1008$, $p<0.05$). **D**) While there were more males from infected lines, comparison of transmission efficiency between infected parents denotes that offspring are more likely to inherit fungus when mothers were infected (Mann–Whitney test: $p = 0.0213$).

## Establishment of *Leptosphaerulina* sp. activates stress and host innate immunity

In response to fungus infection, we observed a systemic immune response in *Anopheles gambiae*. Deposition of melanin in *Leptosphaerulina* sp. infected tissues persisted across developmental stages (Fig 4A–4C). We hypothesized this could elicit a potent systemic immune

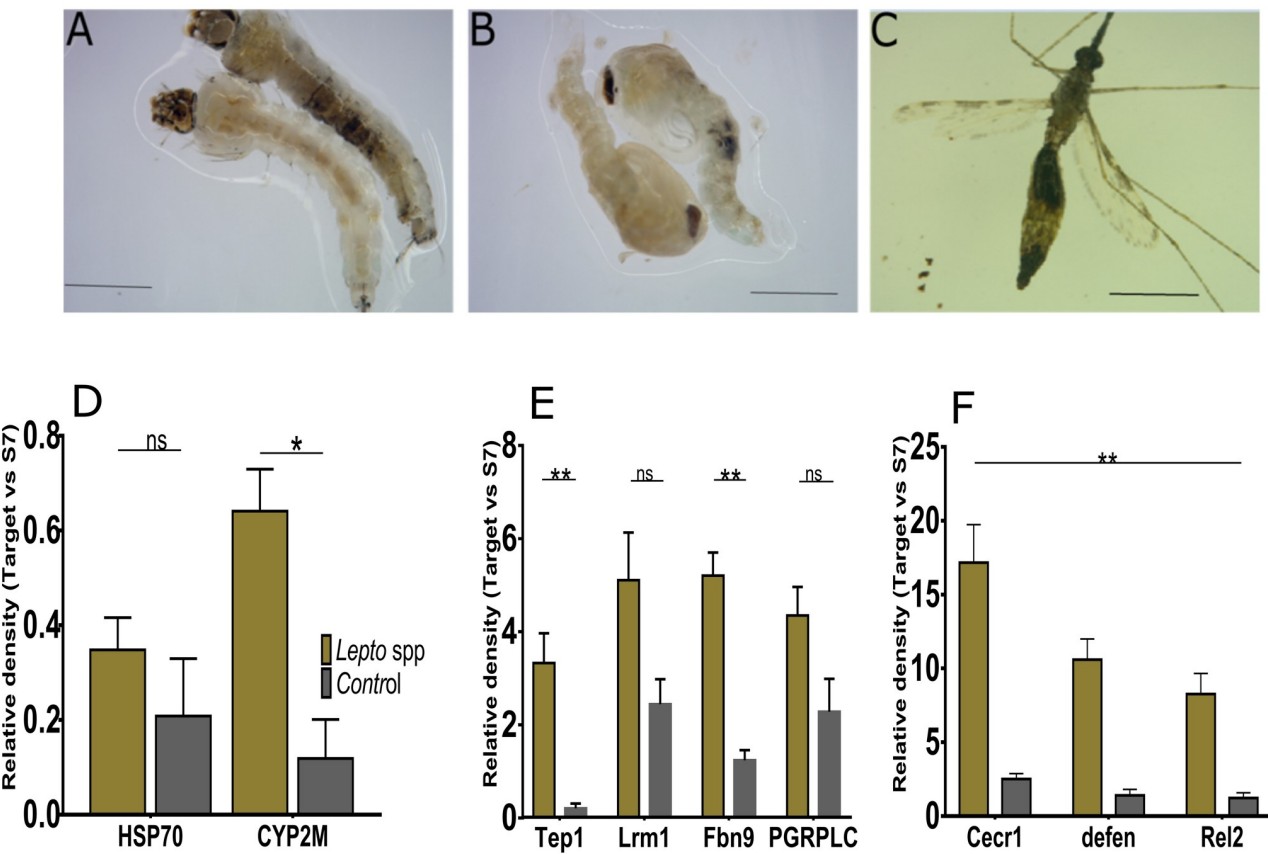

**Fig 4. *Leptosphaerulina* sp. infection induces upregulation of stress and immune genes in *An. gambiae* female. A-C)** Distinct melanosis was observed in fungus infected tissues persisting from larvae to adults. Relative densities were measured by qPCR, and are expression as ratios between the target immune gene and the ribosomal host gene. To undertake this experiment, RNA was extracted from dissections of adult females aged 10 days post-eclosion from previously exposed colonies at larval stage (L1). **D)** Genes involved in exenobiotic metabolism were significantly upregulated (*Cyp6m2*) (Mann-Whitney test, p<0.05). **E)** Two of the IMD genes Thioester-containing protein (*Tep1*) and a fibrinogen-related protein (*FBN9*) were significantly upregulated while leucine-rich repeat immune protein (*LRM1*) and peptidoglycan recognition protein (*PGRP-LC*) did not indicate any statistical differences. **F)**. Toll pathway genes cecropin (*Cecr1*), defensin 1 (*defen*), and Relish ortholog (*Rel2*) were significantly upregulated (Mann-Whitney test, p<0.05) Error bars signify mean±SEM, while * represent p <0.05, ** represents p<0.001, ns = not significant (*p*>0.05). Samples were obtained by pooling 5 dissected guts of confirmed positive.

response coinciding with the widespread of infection. Stress gene *Hsp70* previously shown to influence ability of pathogen propagation in *Anopheles* [37] was moderately elevated while gene a involved in xenobiotic metabolism (*Cyp6m2*) was significantly upregulated(Mann Whitney test, *p* = 0.0189; Fig 4D). Notably, genes of IMD pathway namely *Tep1* and *Fbn9* were significantly upregulated(Mann Whitney test, *p*< 0.01; Fig 4E). Changes in the luciferase gene measured by assessing activity of *cecropin 1* promoter, a gene linked to drive its expression [38], and other genes *defensin 1* and *Rel2* were significantly upregulated in fungus-infected tissues (Mann Whitney test, *p*< 0.01; Fig 4F). Our experiments indicate that a 7 days old adult infected at larval stage presented with relatively constant densities of fungus infection (Fig 1E). Besides, we reasoned that at this time point females had attained maturity to actively seek a blood meal. Re-isolation of this fungus through culture and sequencing methods suggest persistent proliferation of *Leptosphaerulina* sp. isolates within host tissues, an observation supported by data showing that *Leptosphaerulina* sp. impose systemic infection with higher affinity for the gut and reproductive tissue (S4C Fig).

## Discussion

Herein, we used *Anopheles gambiae* model to show that *Leptosphaerulina* sp. fungus isolated from host midguts occur naturally at moderate levels in field collected larvae and adults across different geographical locations and on experimental re-introduction to the host a stable association was observed. Specifically, we determined that *Leptosphaerulina* sp. is likely to be ingested by the immature stages of *Anopheles* sp. and undergo transstadial transmission across life history (to adults) and subsequently from adults to offspring. We found that fungus infection causes deposition of melanin leading to systemic immune response, which could be playing a role in host colonization.

Reports document existence of horizontal and transstadial transfer of symbionts from pupae to adult in mosquitoes [14, 30, 32]. It has been shown that some microbes are transstadially transmitted from larvae to adults [12, 31, 39, 40], and that a shift from aquatic to terrestrial leads to midgut renewal that causes microbial reduction or elimination with the retention of essential microbes [24, 41]. Other observations indicate that most symbionts are lost when pupae emerge to adult [23, 31]. A model proposed by Moll et al. [42] suggest that bacterial reduction takes place in the pupae during eclosion to newly emerged adults leading to transfer of fewer microbes from either the emerging water or sugar sources. These reports support our observation on the reduction in prevalence and densities of most fungus species that we studied. Our experiments showed that surface sterilization of eggs with 1% 1MHCl cleared resident fungi and did not slow larval development. We demonstrate that fungal symbiont associates with *An. gambiae* by colonizing the midgut and reproductive tissues, and this suggest a possible transmission of this fungus through horizontal and/or vertical routes.

We observed that *Leptosphaerulina* sp. increased the duration of larvae development, but that did not affect the numbers of larvae arising from infected colonies. Increased larvae development time could be attributed to elimination of other non-fungi microbes since studies have demonstrated the importance of microbes in the development of anopheline hosts [12, 14]. It has been shown that axenic *Aedes* mosquitoes exhibited delayed time of development [43], and this was coupled with increased lifespan of adults, which could explain the lack of difference we observed in adult longevity between the fungus infected and uninfected mosquitoes. Fungus infected mosquitoes displayed a significant decrease in fecundity. It is possible that curing mosquitoes before fungus inoculation eliminated core microbiomes, which have been suggested to influence capacity of females to reproduce [44, 45]. Notably, we did not observe any significant difference in fecundity of the offspring, suggesting that the fitness cost during the Go were not generational.

Isolation of this fungus in field samples from western (12–19%), central (22–27%), and coastal (11–18%) Kenya indicates natural occurrence in wild mosquitoes. Intensities of naturally *Leptosphaerulina*-infected field *An. gambiae* remained relatively constant in field larvae and adults indicated that this fungus is prevalent at low and moderate levels, and were likely to co-exist with these hosts in a natural ecosystem. This supports our observation that it persists in hosts when introduced artificially. Significant lethal effects on fecundity and delayed pupation could have occurred due to clearing of microbes with 1% HCl leading to microbial shift that favored colonization of this isolate manifested by the development of melanosis, and this could have interfered with the host fitness.

The presence of *Leptosphaerulina* sp. in *Anopheles gambiae* induces a major up-regulation of the host immune genes. It was shown previously that commensal microbiota influences the basal immunity in a way to help in the establishment of bacteria symbiont [35], or curb the development of pathogen in the midgut [46, 47]. We reasoned that fungal proliferation may

result in changes to the basal immunity, accounting in part to the mycobiome establishment and persistence along the developmental stage. Changes in the expression profile of gene involved in metabolism of xenobiotics and effector molecules such as *Defensin 1*, *Cecropin 1*, *Fbn9*, *Rel2*, and *Tep1* suggests the possibility that *Leptosphaerulina* fungi infection induces upregulation of a widespread basal immunity. We reasoned that gene upregulation was sufficient to achieve establishment of fungus and its subsequent transmission as previously demonstrated [35]. A *denovo* formation of melanosis in *An. gambiae* by *Elizabethkingia meningoseptica* has been linked to its vertical transmission in host [35]. Observed persistence of melanin in the infected tissues suggests that *Leptosphaerulina* sp. exploits this phenomenon for increased immune boosting and a stable establishment in *An*. gambiae. Taken together, these observations suggest that fungal infection enhances the expression of mosquito immunity and reactive oxygen species, which might have enhanced *Leptosphaerulina* sp. to establish in the host and persist to its subsequent developmental stages.

These findings have important implications for vector control and transmission of malaria by *Anopheles* mosquitoes. We report that fungi associating with mosquitoes are transstadially transmitted across developmental stages and inherited by their progeny. Upon introduction at the larval stage, *Leptosphaerulina* sp. had low fitness cost on *An. gambiae* host. Delayed pupation and low fecundity could reduce vector density thereby lowering biting rates. However, infected individuals that survive are likely to transmit the infection to the progeny and increase the prevalence and burden of natural infection statuses. Notably, observations made in laboratory infection may not be a reflection of infection in the wild where host are inhabited by other microbes. Importantly, *Leptosphaerulina* sp. may provide a model for understanding interaction between fungus-symbiont with its host. In conclusion, the discovery of a cultivable fungal symbiont that co-exists with the main malaria vector *Anopheles gambiae* and characterization of its role in host development indicate that it could be explored for delivering effectors or *P. falciparum* 'transmission-blocking' molecules in a paratransgenesis-based strategy for controlling malaria.

## Materials and methods

### Ethical statement

The mosquito blood feeding protocol (Ref: KEMRI/RES/7/3/1) was approved by the Kenya Medical Research Institute (KEMRI) and reviewed by the Institutional Animal Care and Use Committee (IACUC) at the International Centre of Insect Physiology and Ecology (*icipe*). Mice were maintained in standard animal houses and provided with food and water *ad libitum* and used solely for blood feeding and maintenance of mosquito colonies. The severity of these procedures is mild to moderate and a reduction in numbers of animals utilized ensures adherence to economic protocol of reduction, refinement and replacement of experimental animal.

### Isolations of fungal symbionts from *Anopheles gambiae* midgut

The *Anopheles gambiae* Mbita strain maintained in semi-field conditions (screen-houses) at iTOC-Mbita point were confirmed microscopically and surface sterilized with 70% ethanol. Briefly, adult samples were washed in a series of EtOH and rinsed in two rounds of PBS before undertaking dissection under microscope. Dissected guts were homogenized in 200μL 1XPBS and serially diluted ($10^{-1}$, $10^{-2}$, $10^{-3}$), apportion (100uL) was plating in Sabouraud Dextrose Agar [SDA] (40 g of sucrose, 10 g of peptone, and 15 g of agar in 1 L of double distilled water), while the other half in Potato Dextrose Agar [PDA] (20 g of sucrose, 10 g of peptone, and 20 g of agar in 1L) and Yeast Peptone Dextrose Agar [YPDA] (Yeast extract (1%), glucose (2%), malt extract (1%), peptone (2%), and agar (2%)) (All media supplied by Oxoid Ltd.,

Basingstoke, Hamp- shire, England). These plates were prepared with a cocktail of tetracycline, and chloramphenicol antibiotics (40μg/mL) and incubated aerobically at room temperature (28.5± 1.5˚C). Pure isolates were maintained in glycerol and water stocks at the Martin Lus- cher Emerging and Infectious Diseases Laboratory (ML-EID) facility at *icipe*.

## DNA sequencing of fungal isolates

Fungi broths were pelleted by centrifuged for DNA extraction. Cells were extracted using pro- tein precipitation method (Puregene, Qiagen, Netherlands). The rRNA internal transcribed spacer (ITS) was amplified using ITS1/ITS4 primers as previously described [36] (sequences in S1 Table). Briefly, the amplification was achieved in a 30 μL reaction mixture containing 19.5 μL PCR $H_2O$, 6μLSolis BioDyne—5x FIREPol$^{®}$ Blend Master Mix—with 7.5 mM MgCl2, 0.75 μL of each primer and 3 μL of DNA template. The mixture was incubated in a SympliAmp PCR machine using the following combinations: initial denaturation at 95˚C for 15min, fol- lowed by 35 cycles of denaturation at 95˚C for 30 s, 58˚C for the annealing of primers at 30 s, 40 s of extension at 72˚C, and a final chain elongation step of 72˚C for 10min. Aliquots of amplicons (5μL) were resolved on a 2% agarose gel stained with ethidium bromide for visuali- zation on a UV-transilluminator. Remaining PCR products with qualified amplification were cleaned with Exosarp IT (USB Corporation, Cleveland, Ohio, USA) according to the manufac- turers' instructions in preparation for sequencing (Macrogen Co., Ltd., South Korea).

## Molecular identification of fungi

Identification of sequences based on their close relatives from the GenBank (http://blast.ncbi. nlm.nih.gov) was undertaken. Multiple alignment of sequences were performed using Muscle [48], with gaps treated as missing data, the tree was constructed using maximum-likelihood method on a Geneious Software v8.1.9. Gene sequences of isolates reported in this work were deposited in GenBank under accession number (see Table 1: MT437213-MT437220; MT433814; MT435649).

## Fungal growth conditions

Fungi isolates from the glycerol stock were revived by transferring stocks from -80˚C, to -20˚, and 4˚C, at 3 hrs interval before taking them to room temperature and inoculating in the PDA, SDA or YPD media to maximize growth. All eight pure isolates were maintained in solid agar and liquid medium. The solid agar consisted of yeast extract (1%), glucose (2%), malt extract (1%), peptone (2%), and agar (2%), while the liquid medium consisted of yeast extract (0.5%), glucose (1%), malt extract (0.5%), and peptone (1%) pH 5.7(Oxoid Ltd., Basingstoke, Hampshire, England). These cultures were maintained at room temperature (28.5±1.5˚C) and monitored for growth. Confirmatory tests were based on colony morphology and qPCR using ITS primer sets (S1 Table).

## Preparation of fungal mycelium for oral introduction

Fungal mycelia were obtained from the individual plates using 5mL double distilled water with a sterile forceps into a pre-weighed 15mL centrifuge tubes. Briefly, the isolates were suspended in distilled water, transferred to the centrifuge tubes and washed by centrifugation at 16500 rpm for 15 minutes to remove traces of media. This was done in three repeats to ensure com- plete drainage of media from the isolates. The pellets were tapped to break the compactness at the bottom of the tube. They were dried at 30˚C in the oven, cooled down to room tempera- ture before taking the final weight to estimate amount of fungi spores per miles. A mile of

vortexed sample was serially diluted and cells counted using haemocytometer, while remaining portion was stored in 4°C awaiting inoculation.

## Curing mosquito eggs and infecting larvae/adults with fungal isolates

Mosquito eggs were cleaned from fungi by washing with 1% HCl and rinsed with double distilled water. This was repeated for two generation when larvae were deemed cured if amplification turned negative with specific ITS1/ITS4 and Lepto primers as listed (S1 Table). Pupae were surface sterilized by washing in four series of distilled to remove any fungus contaminating their surfaces before placing them in the pupal water (25mL) that was acidified by lacing with 250uL hydrochloric acid (1% HCl) and 2.5mL of ketaconazole (20µg/mL). Premature larvae (1$^{st}$ instar—L1) were pre-tested for presence of fungi using ITS1/4 primers as previously described [36]. Colonies that turned negative were infected by inoculating with a one-off dose of 60mg (3.9 x 10$^3$–5.5 x 10$^5$spores/mL) of ground mycelium and spores in triplicates of 60 larvae per standard trough. The controls were maintained on Tetramin™ baby fish food previously tested with ITS1/ITS4 primers to avoid cross contamination. Rearing water was changed every 48 hours interval subsequently. Larvae sample (6–8 days), pupae (10–13) and adults (12–15) were collected to check the presence of fungus using culture and qPCR methods. These were also examined microscopically for any phenotypic changes. Homogenate of dissected midgut and reproductive organs or whole mosquito were plated and half used for DNA analysis. Proportions detected by culture method reflects the number of larvae or pupae and adults for which at least one fungal colony was detected in culture and confirmed by qPCR. For adult infection, newly emerging adults from uninfected lines were allowed to feed on fine whole fungus mixed with 6% glucose solution soaked in cotton wool. However, introduction of fungi to adult was considered inappropriate for testing transmission to offspring since we could not control surface contamination during ovipositing. Contrary, larvae infection could be controlled from contamination to adult stage. To ensure adults did not acquire fungal infection by imbibing pupae water, pupae were surface washed in a series of distilled water laced with 1% 1M Hydrochloric acid (HCl) and transferred to a clean container of distilled water laced with ketaconazole. Adults emerging were immediately aspirated, transferred to a new cage and fungus infection confirmed *post hoc*. In a separate experiment, inoculation of *Leptosphaerulina sp* to uninfectedL1 larvae of *Anopheles gambiae* was used to address whether it exerts adverse or beneficial effects on the host physiology by assaying standard fitness parameters such as proportion of larvae that pupates, proportion of emergence, adult longevity and fecundity. The adults were kept in a room temperature maintained at 28.5±1.5°C and 68–80% relative humidity with a cycle of 12 hrs day (light) and 12hrs night darkness. Controls were fed with 6% glucose soaked in cotton.

## Fitness and transmission assay

We studied larvae fitness by determining the percent survival and the duration taken to pupate, while adult fitness was based on the number surviving over time in comparison to the unexposed mosquitoes. Fungal infected colonies from L1 infection and uninfected virgin mosquitoes were mated with the opposite sex counterparts ($^+$♂x$^+$♀; $^-$♂x$^+$♀; $^+$♂x$^-$♀; $^-$♂x$^-$♀, +signify infected colony while—signify uninfected lines) by allowing them stay in the same cage maintained in a conventional laboratory rearing condition. A healthy BALB/c mouse was placed in each cage of about 40 females to enable blood feeding for 45 minutes under darkness. Engorged females were transferred to individual cups supplied with wet filter papers for egg laying and monitored daily. Filter papers with eggs were removed, eggs counted and immersed in individual hatching trays. Hatchlings or eggs were scored to estimate fecundity for each

mating combination. Females that laid eggs were screened *post-hoc* to ascertain infection status. The number of eggs in each combination was compared with the fungus uninfected females. Fecundity was also analyzed for the offspring (F1 generation) and their infection status confirmed *post hoc*. These experiments were also used to ascertain occurrence of paternal, maternal, or mixed transmission. Fungus density was compared between female and males as well as dissections of midgut and reproductive organs using qPCR assays.

## RNA isolation, first strand cDNA synthesis and quantitative qPCR analysis

A 7 days old adult *Anopheles gambiae* from infected and uninfected lines were dissected to obtain midgut tissues and three individuals pooled for extraction of total RNA using TRIzol reagent (Invitrogen, Carlsbad, CA) according to the manufacturer's instructions. Extracted RNA was treated with DNase I (Ambion) to degrade residual DNA. FIREScript RT cDNA synthesis kit (Solis BioDyne) was used for cDNA synthesis using supplied oligo primers following the manufacturers' instructions. Gene expression levels were measured by running the cDNA in duplicate to perform HRM-qPCR using 5× HOT FIREPol®EvaGreen qPCR Mix Plus (no ROX) as described in the manufacturer's instructions. Gene expression was calculated as the ratio of target gene and ribosomal host gene. All cDNA reagents were procured from Solis Bio-Dyne (Estonia). All the primer sequences used for qPCR are as shown (S1 Table).

## Determining relative density

Relative density of fungi was obtained using primers generated from whole genome sequences relative to host ribosomal S7 protein. Briefly, DNA was extracted using protein precipitation method (Puregene, Qiagen, Netherlands). To undertake HRM-qPCR, a 10uLmastermix was prepared by adding 2uL of HOT FIREPol®EvaGreen® HRM (without ROX), whose components included 5x EvaGreen® HRM buffer, HOT FIREPol® DNA polymerase, dNTPs, BSA, EvaGreen® dye, 12.5 mM Magnesium chloride), 0.5ul of 10 pmol/ul forward and reverse primers (S1 Table), 2 ul of DNA template, and topped up to 10 ul UltraPure nuclease free water (Invitrogen, UK). The PCR amplification program included initial denaturation at 95°C for 15 minutes, 35 cycles of denaturation at 95°C for 30 seconds, annealing at 60°C for 30 seconds, and extension at 72°C for 45 seconds. A final elongation at 72°C for 10 minutes was included before melting the products from 65°C to 95°C at a rise in 1°C interval. The threshold ct value for positive fungal infection was set at 31.0 based on qPCR amplicons resolved on a 2% agarose gel forming a visible band.

## Statistical analysis

A two-tailed test (Mann-Whitney U test) and Kruskal-Wallis tests were used for non-normal distributed data. Correlation coefficient and level of significance were determined from a linear regression test of densities between Go mothers and their offspring while Log-Rank test used for assessing significant levels of the survival data. All analysis was performed in Graph-Pad Prism 7.01 for Windows (GraphPad Software, San Diego, USA). Data are presented as Mean ± SEM of experimental replicates. A *p* value of less than 0.05 was considered statistically significant.

## Supporting information

**S1 Fig. Establishment of fungus in *Anopheles gambiae*.** (**A**) A graphical representation of fungus introduction to anopheles immature stages. To establish fungus free mosquitoes for infection with *Leptosphaerulina* sp., insectary eggs from adults harboring diverse fungi (**B**,

panel **W** screened with ITS1/ITS4 primers expected size 580bp) were washed with 1% HCl for two generations and confirmed fungus negative with ITS1/4 primers (**B**, panel **X**). Uninfected mosquitoes were infected and confirmed to harbor *Leptosphaerulina* sp. at larvae (*L*), pupae (*P*), and adult (*A*) developmental stages using ITS1/4 primers (**B**, panel **Y**, culture isolate [CL] included as positive control while uninfected included as controls). These were confirmed as *Leptosphaerulina* sp. by rescreening infected and uninfected samples using Lepto521F/896R primers developed from isolate whole genome sequence (**B**, panel **Z** with expected size of 320bp). (**C**) The rate of emergence were relatively lower in fungus infected *Anopheles gambiae*. (DOCX)

**S2 Fig. Morphology of *Leptosphaerulina* sp fungus isolated from semi-field mosquitoes.** The isolate is characterized by white colonies (viewed from culture surface) that form septate-hyphae and spore when stained. Pure isolates obtained from sub-cultured isolates of the mid-gut dissection were maintained on Saboroud Dextrose Agar media (**A** is top view of culture on a plate, **B** is the bottom view). Cultures were stained in Lactophenol blue stain for morphological visualization under microscope (**C**).
(DOCX)

**S3 Fig. Variation of *Leptosphaerulina* sp. densities in field caught samples from Mwea location.** Relative densities of *Leptosphaerulina sp* in field Mwea samples at larvae (**A**, consisting of 25 pools of five) and adult (**B**, made of 12 pools of 5) stages were assayed using qPCR-HRM and expressed as the ration of Lepto521f/896r fungal gene against host ribosomal gene.
(DOCX)

**S4 Fig. *Leptosphaerulina* sp. infection on *An. gambiae* F1 progeny.** (**A**) Persistence of the isolate in F1 females did not lower their fertility (Kruska-Wallis test: $p = 0.291$). (**B**) Fungus densities in the F1 infected female and male whole mosquitoes' offspring were assayed by qPCR in 10 day old offspring, and it was noted that intensity of infection burden was relatively high in female (Mann–Whitney test = 1054, *p = 0.0003*). (**C**) These densities were distributed in major tissues with increased burden in the midgut (Kruskal-Wallis test, $p = 0.0056$). The mean bar represents mean±S.E.M., while ** represent *p* value of *<0.01*, *** *p <0.001*, and **** *p < 0.0001*, ns = not significant ($p > 0.05$).
(DOCX)

**S1 Table. List of oligonucleotide primers utilized in the amplification experiments.** (DOCX)

## Acknowledgments

The authors acknowledge Milcah Gitau of *icipe* Arthropod Rearing and Containment Unit for mosquito rearing assistance. We thank Faith Kyengo, Ulrike Fillinger, and Oscar Mbare for assistance and advice. We are also grateful to the African Union under the auspice of Pan African University, Institute for Basic Sciences Technology & Innovation (PAUSTI) for providing postgraduate scholarship.

## Author Contributions

**Conceptualization:** Godfrey Nattoh, Jeremy Keith Herren.

**Data curation:** Godfrey Nattoh.

**Formal analysis:** Godfrey Nattoh.

**Funding acquisition:** Joel L. Bargul, Gabriel Magoma, Jeremy Keith Herren.

**Investigation:** Godfrey Nattoh.

**Methodology:** Godfrey Nattoh, Lilian Mbaisi, Hellen Butungi, Enock Mararo, Evan Teal.

**Project administration:** Gabriel Magoma.

**Resources:** Godfrey Nattoh, Joel L. Bargul, Jeremy Keith Herren.

**Supervision:** Joel L. Bargul, Gabriel Magoma, Jeremy Keith Herren.

**Validation:** Godfrey Nattoh.

**Visualization:** Godfrey Nattoh.

**Writing – original draft:** Godfrey Nattoh, Jeremy Keith Herren.

**Writing – review & editing:** Godfrey Nattoh, Joel L. Bargul, Gabriel Magoma, Lilian Mbaisi, Hellen Butungi, Enock Mararo, Evan Teal, Jeremy Keith Herren.

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
