## [Decision Letter · Decision Letter 0]

1 Oct 2020

PONE-D-20-26091

A native mosquito fungus Leptosphaerulina sp exploits melanotic phenotypes to persist in Anopheles gambiae

PLOS ONE

Dear Dr. Nattoh,

Thank you for submitting your manuscript to PLOS ONE. After careful consideration, we feel that it has merit but does not fully meet PLOS ONE’s publication criteria as it currently stands. Therefore, we invite you to submit a revised version of the manuscript that addresses the points raised during the review process.

1) Reorganize the introduction and discussion emphasizing the host-microbe theoretical framework and reducing the discussion section.

2) We suggest a test of survival will be better than thus U-test. Authors can use Log-rank or Gehan tests, with Kaplan-Meier cumulative survival.

We look forward to receiving your revised manuscript.

Kind regards,

Humberto Lanz-Mendoza

Academic Editor

PLOS ONE

Journal Requirements:

2. In your Methods, please include full details of the housing and care of the mice used to provide mosquito blood meals.

Reviewers' comments:

Reviewer's Responses to Questions

**Comments to the Author**

1. Is the manuscript technically sound, and do the data support the conclusions?

Reviewer #1: Yes

Reviewer #2: Partly

2. Has the statistical analysis been performed appropriately and rigorously? 

Reviewer #1: Yes

Reviewer #2: No

3. Have the authors made all data underlying the findings in their manuscript fully available?

Reviewer #1: Yes

Reviewer #2: Yes

4. Is the manuscript presented in an intelligible fashion and written in standard English?

Reviewer #1: Yes

Reviewer #2: Yes

5. Review Comments to the Author

Reviewer #1: Dear authors,

Your paper is very interesting because it shows a rare study case: the contribution of fungus that are transmitted vertically and horizontally on the insect fitness. I liked the paper. I just have some questions and precisions on the outcome of your study.

Line 28. Please provide a general introduction about biological interactions, so that your writing will not only be interesting for researchers involved in mosquitoes, but for a more general audience. The same in discussion. For example, you can mention that most work have been focused on the contribution of bacteria and less on fungi.

Lines 31-37. Specifically, how your study contributes to resolve these problems? I suggest you to delete this information unless you have a real goal to justify your work. You can justify your work just with basic biology of interactions.

Line 43. Please be more specific. Do you mean that there is no information about whether such isolates are baneful or impose fitness cost on mosquitoes?

Lines 47-48. Again, I did not see how a basic study of interactions contribute to eradicate or diminish malaria. Please be more specific or delete these lines.

Line 82. Please introduce much better the terms melanosis and provide examples.

Line 136. If you have the data, a test of survival will be better than thus U-test. You can try with Log-rank or Gehan tests, with Kaplan-Meier cumulative survival.

Lines 143-144. If you have significant differences, why is this moderate? A moderate difference is a marginal significant result.

Lines 169-170. Although this seems not to be a significant difference, please carry out a Chi-square test.

Lines 304-306. However, this are significant results, and across generations, may be a a potent natural selective pressure. Your data could be argued as parasitism because you have significant results in terms of egg production, development, survival and increase immune response. Please discuss this.

Line 351-352. However, it had significant differences, not marginal statistical differences. One could argue a real cost within and between generations.

Please discuss better your point of melanosis because it was difficult for me to follow the point.

Why the fungus may disappear in laboratory?

Reviewer #2: This is a very nice piece of research, lots of good work. I have a few constructive comments that I invite authors to pay attention:

1. The onset and lay out of the Introduction (and the end of the Discussion) is directed towards strategies of mosquito control. I think this emphasis is way too premature for the type of findings presented. There is nothing wrong with simply describing the same experiments using a host-microbe theoretical framework for example. Otherwise you are misleading your readers.

2. Related to the above comment, it is not so straightforward to me the use of your findings for mosquito control as discussed in the last lines of your discussion. My humble opinion is that you are overselling your ms.

2. The Discussion is way too long.

3. The Results section include several statements that must be placed in the M&M sections (eg. 102-106, 109-111, 131-134 to provide just a few cases).

4. I wonder whether you tried to transform your data and thus avoid so many non-parametrical tests. From the way several graphs look, it seems to me that some sort of transformation is viable.

6. PLOS authors have the option to publish the peer review history of their article (what does this mean?). If published, this will include your full peer review and any attached files.

Reviewer #1: No

Reviewer #2: **Yes: **Alex Córdoba-Aguilar

---

## [Author Response · Author response to Decision Letter 0]

15 Oct 2020

Points raised by the academic editor

1) Reorganize the introduction and discussion emphasizing the host-microbe theoretical framework and reducing the discussion section.

We have reorganized the introduction and discussion parts by putting more emphasis on the interaction between host and microbe. We have also refined the discussion.

2) We suggest a test of survival will be better than thus U-test. Authors can use Log-rank or Gehan tests, with Kaplan-Meier cumulative survival.

It was suggested that we replace U-test with Log-rank because the labeling of intended figure on Y-axis was wrong (Figure 2A). We acknowledge and regret this mistake. We have replaced the label on Y-axis with “proportions of larvae” because it is based on percentage of larvae that developed to pupae and not ‘larvae survival’ as earlier shown. We intended to show that infection with candidate fungi did not induce change in the proportion of larvae developing to pupae. A log-rank test would have been useful in determining the difference between the numbers of larvae (infected and uninfected) surviving over the course of time taken to develop into pupae. However, we used Log-rank to compare survival of infected and uninfected adults (Figure 2D). 

3) Journal Requirements:

We are confident that we have now addressed the vast majority of the journal requirement as provided in the PLOS ONE style template 

4) In your Methods, please include full details of the housing and care of the mice used to provide mosquito blood meals.

We have included full details of housing and care of mice used to provide blood meals. We also provided guiding statement on the severity of this procedure and the principle of reduction in number of the animals used as required in the animal care protocol.

5) Please amend your list of authors on the manuscript to ensure that each author is linked to an affiliation. Authors’ affiliations should reflect the institution where the work was done (if authors moved subsequently, you can also list the new affiliation stating “current affiliation:….” as necessary).

We check authors list and their affiliation and have addressed these points 

Responses to reviewer comments:

Reviewers' comments:

Reviewers (Comments to the Author):

1. Is the manuscript technically sounds and do the data support the conclusions?

Reviewer #1: Yes

We thank the reviewer for this assessment of the implications of our findings

Reviewer #2: Partly

We thank the reviewer for this assessment and confirm that we have since undertaken refinement of our manuscript and linked our finding to host-symbiont interaction. 

2. Has the statistical analysis been performed appropriately and rigorously?

Reviewer #1: Yes

We thank the reviewer for this assessment of the implications of our findings

Reviewer #2: No

We selected methods of data analysis based on the normality test results of our raw data. We didn’t see any need of transforming non-normal data that we could compare mean ranks using U-test or Kruskal-Wallis test. We confirm that the p values we obtained on transformed data were different from those obtained with non-transformed data, but their statistical significance was not different.

3. Have the authors made all data underlying the findings in their manuscript fully available?

Reviewer #1: Yes

We thank the reviewer for this assessment

Reviewer #2: Yes

We thank the reviewer for this assessment

4. Is the manuscript presented in an intelligible fashion and written in standard English?

Reviewer #1: Yes

We thank the reviewer for this assessment

Reviewer #2: Yes

We thank the reviewer for this assessment

5. Review Comments to the Author

Reviewer #1:

Your paper is very interesting because it shows a rare study case: the contribution of fungus that are transmitted vertically and horizontally on the insect fitness. I liked the paper. I just have some questions and precisions on the outcome of your study.

We thank the reviewer for this assessment of the implications of our findings

Line 28. Please provide a general introduction about biological interactions, so that your writing will not only be interesting for researchers involved in mosquitoes, but for a more general audience. The same in discussion. For example, you can mention that most work have been focused on the contribution of bacteria and less on fungi.

We have reorganized the introduction and discussion to emphasize on the host-microbe interaction and paid attention to immense work on host-bacteria with few studies on host-fungi interaction. We believe this will be of interest for a general audience.

Lines 31-37. Specifically, how your study contributes to resolve these problems? I suggest you to delete this information unless you have a real goal to justify your work. You can justify your work just with basic biology of interactions.

We acknowledge that our work does not provide sufficient data to support these claims and have since focused our study on host-symbiont interaction.

Line 43. Please be more specific. Do you mean that there is no information about whether such isolates are baneful or impose fitness cost on mosquitoes?

We confirm that we have provided specific examples of non-entompthogenic fungi symbionts and reported roles of these isolates on fitness cost on mosquitoes

Lines 47-48. Again, I did not see how a basic study of interactions contribute to eradicate or diminish malaria. Please be more specific or delete these lines.

We have reframed the sentence to include “ideal symbiont candidate for the delivery of anti-plasmodium molecules in paratransgenic approach” we believe this symbiont can be modified with anti-plasmodium effectors molecules and reintroduced in these hosts thereby utilized in a paratransgenesis.

Line 82. Please introduce much better the terms melanosis and provide examples.

We have included a description and causes of melanin and two references were added in support (ref. 33, and 34)

Line 136. If you have the data, a test of survival will be better than thus U-test. You can try with Log-rank or Gehan tests, with Kaplan-Meier cumulative survival.

We made a mistake in labeling this figure (Figure 2A). The label on Y-axis should be replaced with “proportions of larvae” because it is based on larvae that developed to pupae. We intended to show that the infection with candidate fungi did not induce change in the proportion of larvae developing to pupae. A log-rank test would have been useful in determining the difference between the numbers of larvae surviving on the course of time taken to develop into pupae.

Lines 143-144. If you have significant differences, why is this moderate? A moderate difference is a marginal significant result.

We have replaced moderate with significant difference.

Lines 169-170. Although this seems not to be a significant difference, please carry out a Chi-square test.

We have carried out a chi-square test and found no significance difference, and included this test results in our MS

Lines 304-306. However, this are significant results, and across generations, may be a a potent natural selective pressure. Your data could be argued as parasitism because you have significant results in terms of egg production, development, survival and increase immune response. Please discuss this.

We have replaced moderate with significant difference and argued that clearing of native microbes with 1% HCl before infection with candidate symbiont could have created microbial shift and favored colonization of the this isolate thereby interfering with the host fitness.

Line 351-352. However, it had significant differences, not marginal statistical differences. One could argue a real cost within and between generations.

We observed significance fitness cost (delayed pupation and fecundity) within generation, but did not find any deleterious effects across generation (fecundity was not statistical significant different) 

Please discuss better your point of melanosis because it was difficult for me to follow the point.

We have included a discussion on melanosis, citing examples, and how it relates with our observed finding (ref. 33 and 34)

Why the fungus may disappear in laboratory?

We had sampled field mosquitoes from wide geographical locations and found this isolate to religiously co-exist with wild mosquitoes at low to moderate levels (densities were lower than those obtained by laboratory infection). We believe a reduction in the laboratory infected individuals could be occurring because we are using germ-free individuals that lack native microbes and since most of the heavily infected mosquitoes will not grow fast, emerge, or produce more eggs there is chance for a reduction but not ‘disappearance’ in the laboratory.

Reviewer #2: 

This is a very nice piece of research, lots of good work. I have a few constructive comments that I invite authors to pay attention:

We thank the reviewer for this assessment of the implications of our findings

1. The onset and lay out of the Introduction (and the end of the Discussion) is directed towards strategies of mosquito control. I think this emphasis is way too premature for the type of findings presented. There is nothing wrong with simply describing the same experiments using a host-microbe theoretical framework for example. Otherwise you are misleading your readers.

We have revised the introduction and discussion parts of the MS putting more emphasis on host-microbe interaction and agree with the reviewer’s observation that the current finding does not provide a direct strategy for parasite/mosquito control.

2. Related to the above comment, it is not so straightforward to me the use of your findings for mosquito control as discussed in the last lines of your discussion. My humble opinion is that you are overselling your ms.

We have tuned down the last line of discussion to focus on host-microbe association.

2. The Discussion is way too long.

We have revised the discussion and shortened it to the best of our ability

3. The Results section include several statements that must be placed in the M&M sections (eg. 102-106, 109-111, 131-134 to provide just a few cases).

We have revised the MS and transferred all statements that describe procedures utilized in M&M sections

4. I wonder whether you tried to transform your data and thus avoid so many non-parametrical tests. From the way several graphs look, it seems to me that some sort of transformation is viable.

We did transform data on egg laid and ran unpaired t-test and found a statistical value of p=0.015. While this value is still lower that 0.05, a previous U-test that took into account individuals that did not lay eggs had reported a statistical value of p=0.0047. It is important to note that we checked whether our data met normality test based on Shapiro-Wilk normality test. Where data were non-normal we decided to use either a two-tailed Mann-Whitney U test or Kruskal-Wallis test.

---

## [Decision Letter · Decision Letter 1]

20 Jan 2021

The fungus Leptosphaerulina persists in Anopheles gambiae and induces melanization

PONE-D-20-26091R1

Dear Dr. Nattoh,

We’re pleased to inform you that your manuscript has been judged scientifically suitable for publication and will be formally accepted for publication once it meets all outstanding technical requirements.

Kind regards,

Tushar Kanti Dutta, Ph.D.

Academic Editor

PLOS ONE

Additional Editor Comments (optional):

Reviewers' comments:

Reviewer's Responses to Questions

**Comments to the Author**

1. If the authors have adequately addressed your comments raised in a previous round of review and you feel that this manuscript is now acceptable for publication, you may indicate that here to bypass the “Comments to the Author” section, enter your conflict of interest statement in the “Confidential to Editor” section, and submit your "Accept" recommendation.

Reviewer #1: All comments have been addressed

Reviewer #2: All comments have been addressed

2. Is the manuscript technically sound, and do the data support the conclusions?

Reviewer #1: (No Response)

Reviewer #2: Yes

3. Has the statistical analysis been performed appropriately and rigorously? 

Reviewer #1: (No Response)

Reviewer #2: Yes

4. Have the authors made all data underlying the findings in their manuscript fully available?

Reviewer #1: (No Response)

Reviewer #2: Yes

5. Is the manuscript presented in an intelligible fashion and written in standard English?

Reviewer #1: (No Response)

Reviewer #2: Yes

6. Review Comments to the Author

Reviewer #1: (No Response)

Reviewer #2: Thanks very much for dealing with my comments so diligently. One last observation. I suggest authors read the paper: Jiménez-Cortés, J. G., García-Contreras, R., Bucio-Torres, M. I., Cabrera-Bravo, M., Córdoba-Aguilar, A., Benelli, G., & Salazar-Schettino, P. M. (2018). Bacterial symbionts in human blood-feeding arthropods: patterns, general mechanisms and effects of global ecological changes. Acta tropica, 186, 69-101. This very recent source can be used and cited for those lines where symbiont and insect vectors interactions are implied (e.g. lines 33-36, 286-288).

7. PLOS authors have the option to publish the peer review history of their article (what does this mean?). If published, this will include your full peer review and any attached files.

Reviewer #1: No

Reviewer #2: **Yes: **Alex Córdoba Aguilar

---

## [Editor Report · Acceptance letter]

12 Feb 2021

PONE-D-20-26091R1 

The fungus *Leptosphaerulina* persists in *Anopheles gambiae* and induces melanization 

Dear Dr. Nattoh:

I'm pleased to inform you that your manuscript has been deemed suitable for publication in PLOS ONE. Congratulations! Your manuscript is now with our production department. 

Kind regards, 

on behalf of

Dr. Tushar Kanti Dutta 

Academic Editor

PLOS ONE